# The Impact of Retrospective Childhood Maltreatment on Eating Disorders as Mediated by Food Addiction: A Cross-Sectional Study

**DOI:** 10.3390/nu12102969

**Published:** 2020-09-28

**Authors:** Rami Bou Khalil, Ghassan Sleilaty, Sami Richa, Maude Seneque, Sylvain Iceta, Rachel Rodgers, Adrian Alacreu-Crespo, Laurent Maimoun, Patrick Lefebvre, Eric Renard, Philippe Courtet, Sebastien Guillaume

**Affiliations:** 1Department of Psychiatry, Saint Joseph University-Hôtel Dieu de France Hospital, Beirut BP 166830, Lebanon; sami.richa@usj.edu.lb; 2PSNREC, University of Montpellier, INSERM, CHU de Montpellier, 34295 Montpellier, France; maude.seneque@gmail.com (M.S.); adrian.alacreu@uv.es (A.A.-C.); philippecourtet@gmail.com (P.C.); s-guillaume@chu-montpellier.fr (S.G.); 3Clincal Research Center and Department of Cardiac and Thoracic Surgery, Saint Joseph University-Hôtel Dieu de France Hospital, Beirut BP 166830, Lebanon; ghassan.sleilaty@usj.edu.lb; 4Department of Psychiatric Emergency and Acute Care, Lapeyronie Hospital, CHRU, 34295 Montpellier, France; rodgers.rachel@gmail.com; 5Quebec Heart and Lung Institute (IUCPQ), Québec, QC G1V 4G5, Canada; sylvain.iceta.1@ulaval.ca; 6School of Nutrition, Laval University, Québec, QC G1V 0A6, Canada; 7Referral Center for Eating Disorder, Hospices Civils de Lyon, F-69677 Bron, France; 8APPEAR, Department of Applied Psychology, Northeastern University, Boston, MA 02115, USA; 9PhyMedExp, University of Montpellier, INSERM, CNRS, 34295 Montpellier, France; l-maimoun@chu-montpellier.fr; 10Département de Médecine Nucléaire, Hôpital Lapeyronie, Centre Hospitalier Régional Universitaire (CHRU) Montpellier, 34295 Montpellier, France; 11Department of Endocrinology, Diabetes, and Nutrition, CHRU, 34295 Montpellier, France; p-lefebvre@chu-montpellier.fr (P.L.); e-renard@chu-montpellier.fr (E.R.); 12UMR CNRS 5203, INSERM U1191, Institute of Functional Genomics, University of Montpellier, 34295 Montpellier, France

**Keywords:** eating disorders, food addiction, childhood trauma, maltreatment, physical neglect

## Abstract

Background: The current study aimed to test whether food addiction (FA) might mediate the relationship between the presence of a history of childhood maltreatment and eating disorder (ED) symptom severity. Methods: Participants were 231 patients with ED presenting between May 2017 and January 2020 to a daycare treatment facility for assessment and management with mainly the Eating Disorder Inventory-2 (EDI-2), the Child Trauma Questionnaire (CTQ), and the Yale Food Addiction Scale (YFAS 2.0). Results: Participants had a median age of 24 (interquartile range (IQR) 20–33) years and manifested anorexia nervosa (61.47%), bulimia nervosa (16.88%), binge-eating disorders (9.09%), and other types of ED (12.55%). They were grouped into those likely presenting FA (*N* = 154) and those without FA (*N* = 77). The group with FA reported higher scores on all five CTQ subscales, as well as the total score of the EDI-2 (*p* < 0.001). Using mediation analysis; significant indirect pathways between all CTQ subscales and the EDI-2 total score emerged via FA, with the largest indirect effect emerging for physical neglect (standardized effect = 0.208; 95% confidence interval (CI) 0.127–0.29) followed by emotional abuse (standardized effect = 0.183; 95% CI 0.109–0.262). Conclusion: These results are compatible with a model in which certain types of childhood maltreatment, especially physical neglect, may induce, maintain, and/or exacerbate ED symptoms via FA which may guide future treatments.

## 1. Introduction

According to the World Health Organization, “childhood maltreatment is the abuse and neglect that occurs to children under 18 years of age. It includes all types of physical and/or emotional ill-treatment, sexual abuse, neglect, negligence, and commercial or other exploitation, which results in actual or potential harm to the child’s health, survival, development, or dignity in the context of a relationship of responsibility, trust, or power” [1]. On the other hand, eating disorders (EDs) are multifactorial mental disorders affecting young individuals and are associated with a mortality rate higher than that of the general population of the same age [2]. The relationship between a history of childhood maltreatment and the later development of an ED is well established, as supported by two major meta-analyses [3,4]. Abused and/or neglected children who have experienced any type of maltreatment (i.e., emotional, sexual, and physical) are at least threefold more likely to develop a future ED [3,4]. Furthermore, a dose–effect relationship between the number of subtypes of childhood trauma experienced and the severity of ED clinical features has been evidenced, suggesting a consistent and partly independent association between these traumatic events and more severe clinical and functional characteristics of ED [5]. While childhood maltreatment may be reported by a high proportion of patients with ED, only a minority of those previously exposed to one or more traumatic events (9–24%) may subsequently present a comorbid post-traumatic stress disorder (PTSD) [6]. Accordingly, beyond the simple comorbidity with PTSD, it is not yet understood how different types of childhood maltreatment impact the clinical presentation of ED, with emotion dysregulation being consistently considered as an important factor mediating this effect [6,7].

In addition to emotion dysregulation, PTSD, and depression as known mediators of ED development in patients who have been exposed to childhood maltreatment, food addiction (FA) may constitute a yet unexplored contributing mediator [6,7,8,9]. FA is characterized by poorly controlled intake of preferred foods, which are postulated to act via similar mechanisms as both illicit and licit drugs of abuse in the brain [10]. An increasing amount of evidence of biological and behavioral changes in response to preferred foods (such as brain reward changes, impaired control, genetic susceptibility, substance sensitization and cross-sensitization, and impulsivity) has been sufficiently convincing to conceptualize FA as an addiction disorder [11]. FA has been increasingly considered as an important psychological dimension that leads, in patients with a history of complex trauma, to ED and more specifically binge-eating disorders (BED) and bulimia nervosa (BN) [10]. Despite being a clinical manifestation of addiction to food, as much as 61.5% of patients with anorexia nervosa (AN) of the restrictive type were found to suffer from FA, which translates how much the addictive behavior related to food can be a common pathological dimension to all EDs, as well as a possible accompanying manifestation of other forms of behavioral addiction to fasting, physical exercising, etc. [12]. 

Although data on the relationships between childhood maltreatment and FA are lacking to date, evidence from clinical studies examining closely related dimensions suggests that such an association might exist. Although not measuring childhood maltreatment per se, in a cohort study of 49,408 female nurse participants, the prevalence of FA increased with the number of lifetime PTSD symptoms, and women with the greatest number of PTSD symptoms reported more than twice the prevalence of FA compared to women with neither PTSD symptoms nor trauma histories. Interestingly, however, in this study, the relationship between FA and PTSD did not differ by trauma type [13]. Furthermore, the co-occurrence of FA symptoms with emotional dysregulation symptoms has led to the suggestion that these might share common characteristics and, potentially, risk factors [14]. In further support of this, when compared to individuals without addictive behaviors, both women with FA and women with substance use presented higher levels of depressive and PTSD symptoms, as well as greater emotion dysregulation [15]. It has been proposed that childhood maltreatment might also be associated with decreased emotional regulation, as well as a greater propensity to and severity of addictive-like behaviors due to structural brain changes (mainly diminished hippocampus volume) [16,17]. Taken together, converging evidence, therefore, seems to exist for an indirect relationship such that FA might constitute an intervening factor in the cross-link between childhood maltreatment and ED symptom severity. 

To our knowledge, no study has yet assessed this proposed indirect pathway in patients with ED. We, therefore, hypothesized that retrospective childhood maltreatment would be indirectly related to ED symptom severity via FA among a transdiagnostic sample of ED patients. In this cross-sectional study design, we aim to establish a conceptual model to further the understanding of how different types of childhood maltreatment might impact ED in order to guide future assessment strategies and treatment development models.

## 2. Materials and Methods

### 2.1. Participants

All consecutive outpatients with all types of ED according to the DSM-5 (Diagnostic and Statistical Manual of Mental Disorders) criteria who were assessed in an eating disorders unit in Montpellier, France, between May 2017 and January 2020 were eligible for the study. Patients with ED are referred to this unit for multidisciplinary assessment, diagnostic confirmation, and management. The data utilized here are drawn from a large study approved by the Ethics Committee of CPP Sud-Est VI of Clermont Ferrand University (CPP: AU 1313; ID-RCB: 2017-A00269-44; N° Clinical Trial: NCT03160443). Signed informed consent was obtained from all participants (and from parents of underage participants). All research procedures were conducted according to the Declaration of Helsinki. Inclusion criteria were as follows: age superior to 15 years (15–70 years), speaking French, and having an ED diagnosis according to DSM-5 criteria. Exclusion criteria were as follows: refusing to consent (*n* = 3), having a mental disability such as intellectual deficiency (*n* = 4), and having a physical comorbidity that prevented study participation (severe hypokalemia that necessitated a transfer to an intensive care unit *n* = 4; very low nutritional state *n* = 9; other physical disorders having an important secondary impact on cognitive functions *n* = 5). 

### 2.2. Measures

The multidisciplinary clinical assessment was carried out during a full day at the outpatient unit by experienced mental health professionals. The ED diagnosis was established on the basis of a nonstructured clinical assessment by psychiatrists, psychologists, and nutritionists, as well as a structured evaluation with the Mini-International Neuropsychiatric Interview (MINI, Version 5.0.0). All investigators were trained beforehand to use the MINI. Body weight and height were collected in a standardized way during the clinical examination. Among other psychometric and biometric assessments, participants completed the questionnaires below.

The Eating Disorder Inventory (EDI-2) is a self-report diagnostic tool designed for use in a clinical setting to assess the clinical dimensions of EDs. It contains 11 subscales (drive for thinness, bulimia, body dissatisfaction, ineffectiveness, perfectionism, interpersonal distrust, interoceptive awareness, maturity fears, asceticism, impulse regulation, social insecurity) that evaluate the symptoms of the ED, as well as its relationship with personality traits and emotions [18,19]. The total EDI-2 score used in this study consists of the sum of all 11 subscales scores [20]. Cronbach’s alpha coefficient for internal consistency was 0.84 for EDI-2 total score, ranging from 0.64 (for the ascetism subscale) to 0.92 (for the bulimic tendency subscale).

The Yale Food Addiction Scale 2.0 (YFAS 2.0) is a 35-item self-report Likert-type scale that assesses food and eating regulation during the past 12 months. Items are scored on an eight-point scale with frequency response options ranging from “never” to “every day.” The items assess clinical impairment/distress according to the DSM-5 criteria for substance use disorder. For the diagnosis of food addiction, the clinically significant impairment/distress criterion has to be met along with two or more diagnostic criteria [21,22]. Cronbach’s alpha coefficient for internal consistency was 0.96 for YFAS 2.0.

The Child Trauma Questionnaire (CTQ) is a 28-item self-report instrument for the retrospective assessment of trauma exposure during childhood. The CTQ consists of five subscales representing different types of trauma (physical abuse, emotional abuse, sexual abuse, physical neglect, and emotional neglect) with multiple items according to a five-point Likert scale ranging from 1 (never true) to 5 (very often true). A higher score on a subscale indicates more severe childhood trauma [23]. Cronbach’s alpha coefficients for internal consistency were 0.89 for the CTQ emotional abuse subscale, 0.94 for the CTQ physical abuse subscale, 0.96 for the CTQ sexual abuse subscale, 0.93 for CTQ emotional neglect, and 0.74 for physical neglect. 

### 2.3. Statistical Analyses

In a first analysis, participants were divided into two groups: those with food addiction (FA(+)) and those without food addiction (FA(−)). Quantitative variables significantly departing from normality assumptions (as assessed by Kolmogorov–Smirnov test and quantile–quantile (Q–Q) plots) were expressed as medians with interquartile ranges (IQR: Q1–Q3). A bivariate comparison between the groups’ characteristics was conducted using the Mann–Whitney U test and Pearson’s chi-square (or Fisher correction as appropriate) test. Cronbach’s alpha was computed for EDI-2 total score, YFAS 2.0, and CTQ subscales.

In a second analysis, the distributions of CTQ subscales, EDI-2 subscales, and YFAS 2.0 were assessed with normality tests (Kolmogorov–Smirnov test and Q–Q plots). Pearson’s correlations were used to estimate index zero-order relationships among childhood maltreatment, FA and ED symptom severity, and their 95% confidence intervals (95% CI) calculated using Yates transform. Mediation analyses examining the hypothesis that food addiction underlies the relationship between childhood maltreatment and ED were tested using the PROCESS Model 4. Bias-corrected bootstrapped confidence intervals (CI) according to 10,000 bootstrap samples were built for the indirect effect (i.e., effect of child trauma (CT) on ED symptoms through FA). FA was considered to exert a mediation effect between childhood maltreatment and ED clinical symptoms when 95% CIs for indirect effects did not overlap with zero [24]. 

## 3. Results

Overall, of the 247 participants assessed, 231 provided YFAS 2.0 data and, accordingly, were included in the study. The majority of participants were women (*n* = 213; 92.2%), with a median age of 24 (IQR 20–33) years, and the most frequent diagnosis was anorexia nervosa (AN) (*n* = 142; 61.47%) followed by BN (*n* = 39; 16.88%), BED (*n* = 21; 9.09%), and other types of ED which were collapsed into a category, including the following DSM-5 diagnoses: (1) avoidant/restrictive food intake disorder; (2) pica; (3) merycism; (4) other specified feeding or eating disorder; (5) unspecified feeding or eating disorder (*n* = 29; 12.55%) (Table 1).

Participants were separated into two groups: 154 (66.66%) with a food addiction (FA(+)) and 77 (33.33%) with no food addiction (FA(−)). The comparison between the FA(+) and FA(−) groups revealed no differences in terms of age, gender, past history of depression, and current diagnosis of AN, BED, and other types of ED. However, FA(+) presented BN (27.5% in FA(+) vs. 8.1% in FA(−); *p* = 0.012) more frequently. Furthermore, actual body mass index (BMI) was higher in the group of patients with FA (20.97 (IQR 16.9–22.1) in FA(+) vs. 17.8 (IQR 16.1–19.9) in FA(−); *p* = 0.005). Moreover, actual and/or past history of PTSD was higher in the FA(+) group (15.58% in FA(+) vs. 2.59% in FA(−); *p* = 0.006). In addition, a current diagnosis of depression was more frequent in patients with FA (36.87% in FA(+) vs. 14.7% in FA(−); *p* = 0.001) (Table 2). In the comparison between both groups, patients in FA(+) presented a higher score on all five subscales of the CTQ. All EDI-2 subscales, as well as EDI-2 total scores, were significantly higher in FA(+) patients except for social insecurity, which was higher in FA(−) patients (*p* < 0.001), and interpersonal distrust, which did not significantly differ between groups (Table 3).

Correlation between CTQ subscales and YFAS 2.0 total score showed a small to moderate effect size with a positive statistically significant correlation with all CTQ subscales, the largest being for emotional abuse (*r* = 0.314; *p* < 0.001) and physical neglect (*r* = 0.307; *p* < 0.001). The YFAS 2.0 scores and EDI-2 total score were positively and significantly correlated with a large effect size (*r* = 0.608; *p* < 0.001). The EDI-2 total score evidenced significant correlations with moderate effect sizes with all CTQ subscales, the highest being for emotional abuse (*r* = 0.349; *p* < 0.001) (Table 4). 

Findings from the mediation analyses are summarized in Table 5, and significant effects are represented in Figure 1. A direct effect between the CTQ subscales and the EDI-2 total score (unmediated by YFAS 2.0) was found for emotional and sexual abuse only (*p* = 0.002 and 0.003, respectively). A consistent indirect mediation effect was present between all CTQ subscales and the EDI-2 total score via YFAS 2.0. The strongest indirect mediation effect was found in relation to the CTQ physical neglect subscale (standardized effect = 0.208; 95% CI 0.127–0.29), followed by emotional abuse (standardized effect = 0.183; 95% CI 0.109–0.262). Accordingly, the mediation or indirect effect of FA related to the impact of childhood maltreatment on clinical symptoms of ED seems to be more specific to physical neglect since, in addition to exerting the highest indirect effect among all CTQ subscales, it did not exert any direct effect on EDI-2 total score (Figure 2).

## 4. Discussion

This study investigated the relationship among self-reported history of childhood maltreatment, FA, and ED symptom severity in a large sample of consecutively recruited patients with ED. Patients with FA reported more frequent histories of childhood maltreatment and presented with more severe ED symptoms as assessed by the EDI-2. In addition, the strong correlation between YFAS and EDI-2 scores suggested that FA and ED symptom severity may be tightly related, while childhood maltreatment was less strongly related to FA. Emotional abuse seems to be the most important type of childhood maltreatment affecting ED symptom severity. In addition, our findings revealed evidence of an indirect effect between all types of childhood maltreatment and ED symptom severity via FA. The strongest of these indirect or mediated effects was related to physical neglect followed by emotional abuse. Moreover, a direct effect relationship between childhood maltreatment and ED symptom severity emerged for the emotional and sexual abuse dimensions. Accordingly, our findings highlight the specific importance of FA in mediating the impact of physical neglect of the clinical severity of patients with any type of ED. Although cross-sectional, these findings are consistent with a model in which retrospective childhood maltreatment, especially physical neglect, might precipitate or constitute a risk factor for FA which may later predispose for, maintain, and/or exacerbate ED symptoms. The role of FA as a mediating factor of in the relationship between retrospective childhood maltreatment and ED warrants further exploration in longitudinal observational studies.

Previous work examined the relationship between trauma exposure and FA. The large cross-sectional cohort study by Mason et al. described above revealed that the likelihood of reporting FA increased with the number of lifetime PTSD symptoms, with the prevalence of FA in women with the greatest number of PTSD symptoms more than twice that of women with neither PTSD symptoms nor trauma histories [13]. Moreover, cross-sectional evidence of relationships between childhood maltreatment and FA and between FA and binge eating was found individuals with higher weight [25]. Further support for this relationship was provided by a comparative study in which individuals with FA reported greater severity of PTSD symptoms as compared to controls [17]. Finally, Stojek et al. revealed that FA severity mediated the association between childhood maltreatment and insulin resistance in women with type 2 diabetes [26]. These cumulative findings, together with ours, suggest a relationship between traumatic exposure and history and FA, as well as other dimensions of disinhibition related to food or substances. Previous work suggested that, at the neurobiological level, both the emotional and the motivational circuits in the brain seem to be affected after exposure to childhood maltreatment [27]. These effects may be associated with disruptions to the experience of inner cues related to the regulation of food and eating, for example, through the effects of stress hormones such as glucocorticoids on the cerebral cortex and limbic system, which may affect the patient’s impulse control [27]. These disruptions and the development of maladaptive behavioral patterns related to food may then increase risk for several types of ED especially those that include bingeing behaviors [28,29]. 

Consistent with this, in our sample, BN was more prevalent in the group of patients reporting FA as compared to the group without FA. This is further in line with findings of relatively recent studies in which FA was found in as many as 96% of patients with BN with a tendency for FA severity to decrease over the course of effective management of BN [30,31]. In a recent study aiming to characterizing FA as a phenotypical construct in patients with different types of EDs and obesity via a factor analysis, results suggested that patients with FA and BN presented with more severe ED psychopathology [32]. Moreover, other work suggested that, among women with a high BMI, the presence of a relationships between early life adversities and FA may be underpinned by specificities in brain regions implicated in reward and emotional regulation [33,34,35]. Similarly, patients with PTSD were described to be at higher risk for FA and EDs due to the potential mediating role of emotional dysregulation [36]. In addition to affecting the activation pattern and connectivity in brain reward circuits, acute and chronic exposure to stress is considered to affect the hypothalamic–pituitary–adrenal axis, leading to multiple pathological cascades that may induce the development of food craving and addiction, as well as symptoms of depression [37,38,39].

The varying findings across the dimensions of childhood maltreatment in our mediation analyses suggest that the relationship between childhood maltreatment and ED symptoms may follow a different pathway depending on the subtype of maltreatment. Indeed, this is consistent with evidence from studies conducted in the past few decades indicating that specific types of childhood maltreatment may be differentially associated with particular types of disordered eating [5,8,40,41,42,43]. Thus, for example, in a large cohort of American young adults, individuals with a history of physical abuse only displayed a higher tendency toward fasting and skipping meals [39]. In contrast, emotional abuse seems to be most consistently related to EDs symptoms, with evidence supporting a mediated pathway via emotional dysregulation [42,43]. In addition, emotional abuse was found to predict higher eating, shape and weight concerns, and poorer functioning in patients with EDs independently of the presence of other comorbidities [5]. Consistent with this, in our study, of the five dimensions of childhood maltreatment assessed, emotional abuse presented the strongest relationship with ED symptom severity. Furthermore, our findings from the mediation analysis revealed that emotional abuse presented the highest total effect of childhood maltreatment on ED symptoms with direct and indirect effects being globally in the same range. The direct effect of emotional abuse in our mediation model predicting ED symptom severity may also reflect the presence of other contributing factors, such as emotional dysregulation, which was not been included here. Further research examining the role of emotional dysregulation in these relationships is warranted. 

Our most interesting finding is related to the fact that the strongest indirect effect of retrospective childhood maltreatment on current ED symptom severity via FA emerged for physical neglect. Moreover, physical neglect’s effect on ED symptom severity was only mediated via FA. Physical neglect refers to the failure to provide a child with basic necessities of life such as food and clothing [44]. Brain maturation via myelination, synaptic plasticity, and the release of neurotransmitters depends largely on the prenatal and postnatal nutritional status of children and adolescents [45]. Indeed, it was shown that parental neglect is an intervening factor in the association between food-approaching appetitive traits and higher weight in children [46]. Accordingly, we can speculate that physical neglect may lead to brain maturation difficulties that may increase risk for FA and, subsequently, an ED. 

Patients with EDs and a history of childhood maltreatment may benefit from care that specifically targets this history [47]. Furthermore, the symptom pathway leading to an ED in individuals with a history of childhood maltreatment has been described as specific to this group. Thus, overvaluation of weight and body shape may lead to feelings of loss of control followed by depressive symptoms and, subsequently, overeating [48]. Tailoring of usual treatment protocols to account for these pathways may help to improve clinical outcomes. Given the evidence found in this study for the mediating role of FA, it would be interesting to evaluate whether, in addition to the usual treatment, therapeutic strategies specifically targeting FA might improve overall prognosis. The presence of a history of physical neglect should raise the clinicians’ index of suspicion for the presence of FA. In this regard, in case FA is confirmed as a comorbid clinical entity accompanying the ED (after using screening tools such as YFAS), the treatment of FA clinical dimensions and its overall impact on ED symptoms should be assessed in future studies. Accordingly, known suggested treatment protocols such as combining pharmacotherapies (opiate antagonists) and psychotherapies (such as cognitive behavioral therapy and psychodynamic group treatments) may be successful in targeting FA clinical dimensions and, subsequently, ED symptom severity [49,50].

The current study includes several limitations. First, all assessments (other than ED diagnosis) were self-reported which might constitute a source of bias. Indeed, participants with more severe clinical dimensions of ED may more easily recall incidents of childhood maltreatment. Second, the study is cross-sectional and retrospective in its assessment of childhood maltreatment, which limits the extent to which the directionality of relationships can be inferred from the findings. Furthermore, the lack of a nonclinical control group of individuals without EDs limited the extent to which confounding factors could be controlled. Finally, EDs were considered as a spectrum of disorders manifesting in different psychopathological dimensions as reflected by the EDI-2 score. However, in the current study only a composite score of ED symptom severity was used, and future work aiming to clarify the relationships among childhood maltreatment, FA, and different dimensions of disordered eating would be valuable. 

## 5. Conclusions

In conclusion, although cross-sectional, our findings support the existence of a mediated relationship between retrospective childhood maltreatment and ED, via FA, especially in the presence of a history of physical neglect. Patients with severe ED symptoms and a history of childhood maltreatment should be systematically assessed for the presence of FA. Moreover, when childhood maltreatment is documented in patients with ED, tailoring treatment plans to specifically address FA should be considered.

## Figures and Tables

**Figure 1 nutrients-12-02969-f001:**
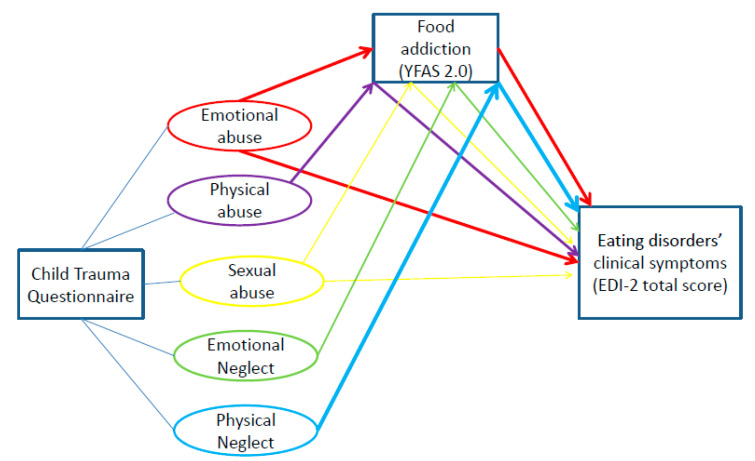
Direct and indirect pathways between childhood maltreatment types and the EDI-2 total score in the mediation analysis. The largest indirect effect emerged for physical neglect (standardized effect = 0.208; 95% CI [0.127-0.29]) followed by emotional abuse (standardized effect=0.183; 95% CI [0.109-0.262]. Arrows width is proportional to the effect size. YFAS 2.0: Yale Food Addiction Scale 2.0; EDI-2: Eating Disorder Inventory-2.

**Figure 2 nutrients-12-02969-f002:**
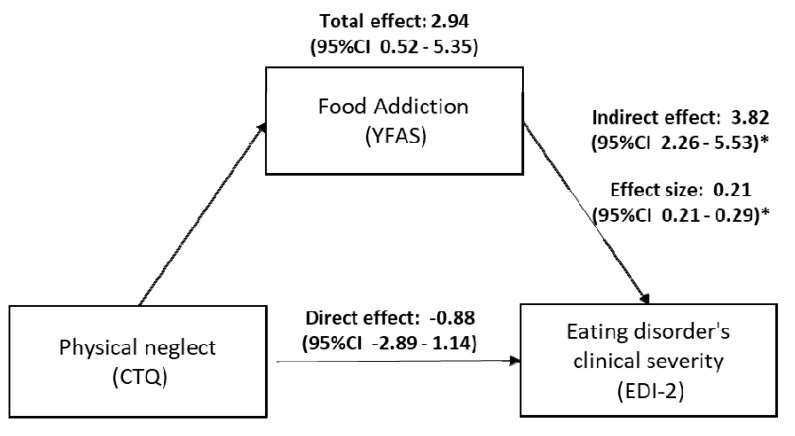
Triangular scheme depicting the results of mediation analysis, with food addiction (measured by YFAS 2.0 score) mediating the effect of physical neglect (CTQ physical neglect component) on eating disorder severity (EDI-2 score). Total, direct, and indirect effects correspond to the beta coefficients obtained from the mediation analysis. The effect size corresponds to the standardized indirect effect of YFAS on EDI-2. 95% CI denotes the 95% confidence interval. YFAS: Yale Food Addiction Scale 2.0, CTQ: Child Trauma Questionnaire, EDI-2: Eating Disorder Inventory-2; (*) confidence limits derived by bootstrapping 10,000 samples.

**Table 1 nutrients-12-02969-t001:** Sociodemographic and clinical parameters, as well as Child Trauma Questionnaire (CTQ) and Eating Disorder Inventory (EDI-2) scores in the entire study population.

Variable	Category	Statistic	All Participants
		*N*	231
Age (years)		Me (IQR)	24 (19–33)
Gender	Male	*N* (%)	18 (7.8%)
	Female	*N* (%)	213 (92.2%)
ED	Diagnosis of AN	*N* (%)	142 (61.5%)
	Diagnosis of BN	*N* (%)	39 (16.9%)
	Diagnosis of BED	*N* (%)	21 (9.1%)
	Other diagnosis	*N* (%)	29 (12.5%)
Current and/or past history of PTSD		*N* (%)	33 (14.3%)
BMI (kg/m²)	Current	Me (IQR)	18.7 (16.8–21.5)
CTQ	Emotional abuse	Me (IQR)	9 (6–13)
	Physical abuse	Me (IQR)	5 (5–7)
	Sexual abuse	Me (IQR)	5 (5–7)
	Emotional neglect	Me (IQR)	12 (8–16)
	Physical neglect	Me (IQR)	7 (5–9)
EDI-2	Drive for thinness	Me (IQR)	22 (17–28)
	Bulimia	Me (IQR)	15 (5–24)
	Body dissatisfaction	Me (IQR)	21 (17–24)
	Ineffectiveness	Me (IQR)	23 (20–27)
	Perfectionism	Me (IQR)	18 (12–22)
	Interpersonal distrust	Me (IQR)	18 (15–20)
	Interoceptive awareness	Me (IQR)	28 (20–34)
	Maturity fears	Me (IQR)	19 (16–22)
	Asceticism	Me (IQR)	19 (13–24)
	Impulse regulation	Me (IQR)	20 (13–28)
	Social insecurity	Me (IQR)	18 (16–21)
	Total score	Me (IQR)	220 (188–254)

ED, eating disorder; PTSD, post-traumatic stress disorder; AN, anorexia nervosa; BN, bulimia nervosa; BED, binge-eating disorder; BMI, body mass index; Me, median; IQR, interquartile range.

**Table 2 nutrients-12-02969-t002:** Comparison between food addiction (FA(−) and FA(+)) groups with regard to sociodemographic parameters, diagnosis, and BMI.

Variable	Category	Statistic	FA(−) Group	FA(+) Group	Test	*p*-Value
		*N*	77	154		
Age (years)		Me (IQR)	28 (19–34)	27.84 (20–32)	U	0.263
Gender	Male	*N* (%)	7 (38.9%)	11 (61.1%)		
	Female	*N* (%)	70 (32.9%)	143 (67.1%)	Chi²	0.603
ED	Diagnosis of AN	*N* (%)	50 (35.2%)	92 (64. 8%)	Chi²	0.339
	Diagnosis of BN	*N* (%)	6 (15.4%)	33 (84.6%)	Chi²	0.012
	Diagnosis of BED	*N* (%)	5 (23.8%)	16 (76.2%)	Chi²	0.352
	Other diagnosis	*N* (%)	7 (50%)	7 (50%)	Chi²	0.172
Current and/or past history of PTSD		*N* (%)	2 (7.7%)	24 (92.3%)	Y	0.006
BMI (kg/m²)	Current	Me (IQR)	17.8 (16.1–19.9)	20.97 (16.9–22.1)	U	0.005

Data are presented as frequency and percentage (*N* (%)) or as median and interquartile range (Me (IQR)). The statistical comparisons in Table 1 were carried out with the Mann–Whitney U test (U), the chi-square test (Chi²), or the Yates test (Y). FA: food addiction.

**Table 3 nutrients-12-02969-t003:** Comparison between FA(−) and FA(+) with regard to CTQ and EDI-2 scores.

		FA(−) Group	FA(+) Group	Test	*p*-Value
	*N*	77	154		
Scale	Subscale				
CTQ	Emotional abuse	7 (5–10)	10 (7–14.25)	U	<0.001
	Physical abuse	5 (5–5)	5 (5–8)	U	0.005
	Sexual abuse	5 (5–5)	5 (5–8)	U	0.014
	Emotional neglect	10 (7–13.75)	13 (9–17)	U	0.005
	Physical neglect	6 (5–8)	7 (6–10)	U	0.006
EDI-2	Drive for thinness	18 (2–22)	25 (10–29)	U	<0.001
	Bulimia	4 (0–14)	19 (2–25)	U	<0.001
	Body dissatisfaction	18 (7–21)	22 (13–25)	U	<0.001
	Ineffectiveness	21 (13–24)	25 (17–29)	U	<0.001
	Perfectionism	15 (4–21)	20 (7–23)	U	0.001
	Interpersonal distrust	17 (11–20)	18 (13–20)	U	0.357
	Interoceptive awareness	21 (6–27)	32 (17–36)	U	<0.001
	Maturity fears	17 (8–20)	19 (13–23)	U	<0.001
	Asceticism	13 (3–18)	21 (8–26)	U	<0.001
	Impulse regulation	13 (2–19)	25 (10–30)	U	<0.001
	Social insecurity	20 (13–23)	18 (13–20)	U	0.001

Data are presented as the median and interquartile range (Q1–Q3).

**Table 4 nutrients-12-02969-t004:** Correlations among Yale Food Addiction Scale (YFAS) total score, EDI-2 total scores, and all CTQ subscales. CI, confidence interval.

CTQ *r* (95% CI); *p*-Value					
	Emotional Abuse	Physical Abuse	Sexual Abuse	Emotional Neglect	Physical Neglect
YFAS	0.314	0.246	0.16	0.208	0.307
*r* (95% CI)	(0.19–0.428)	(0.12–0.365)	(0.028–0.286)	(0.079–0.331)	(0.183–0.421)
*p*-Value	<0.001	<0.001	0.018	0.002	<0.001
0.608					
(0.519–0.684)					
<0.001					
EDI-2	0.349	0.199	0.25	0.227	0.161
*r* (95% CI)	(0.228–0.459)	(0.071–0.322)	(0.121–0.37)	(0.098–0.348)	(0.031–0.287)
*p*-Value	<0.001	0.003	<0.001	<0.001	0.016

**Table 5 nutrients-12-02969-t005:** Analysis of total, direct, and indirect (via YFAS 2.0 mediation) effect of different CTQ subscales on EDI-2 total score.

CTQ Subscales						
Type of Effect						
(CI);						
*p*-Value						
		Emotional Abuse	Physical Abuse	Sexual Abuse	Emotional Neglect	Physical Neglect
	Total effect	3.401	2.433	2.934	2.24	2.944
	(Direct and indirect)	(2.2–4.6)	(0.83–4.03)	(1.42–4.44)	(1–3.49)	(0.52–5.35)
		<0.001	0.003	<0.001	<0.001	0.017
EDI-2						
total score	Direct effect	1.64	0.543	1.809	0.947	−0.873
		(0.6–2.68)	(−0.77–1.85)	(0.6–3.01)	(−0.07–1.96)	(−2.89–1.14)
		0.002	0.417	0.003	0.069	0.396

	Indirect effect	1.761	1.89	1.125	1.303	3.817
		(1–2.66)	(1.04–2.88)	(0.11–2.28)	(0.54–2.16)	(2.26–5.53)
	Standardized indirect effect	0.183	0.153	0.097	0.136	0.208
		(0.1–0.26)	(0.08–0.22)	(0.01–0.18)	(0.05–0.21)	(0.12–0.29)

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
