# Peer review of "The Impact of Retrospective Childhood Maltreatment on Eating Disorders as Mediated by Food Addiction: A Cross-Sectional Study"

_nutrients, 2020, doi:10.3390/nu12102969_

Round 1

Reviewer 1 Report

In the present study, Authors mainly focus on retrospective childhood maltreatment, therefore I would improve the title by “The impact of retrospective childhood maltreatment on eating disorders as mediated by food addiction: a cross-sectional study”.

Introduction

In my opinion comparison between FA and ED (for showing the similarities and discrepancies) will definitely increase the value of the introduction. Perhaps Authors may wish to include some of interesting papers on FA in their introduction:

Gordon, E.L., Ariel-Donges, A.H., Bauman, V., Merlo, L.J. (2018). What is the evidence for “food addiction?” A systematic review. Nutrients, 10(4), 477. https://doi.org/10.3390/nu10040477.

Brewerton, T.D. (2017). Food addiction as a proxy for eating disorder and obesity severity, trauma history, PTSD symptoms, and comorbidity. Eating and Weight Disorders, 22(2), 241-247. https://doi.org/ 10.1007/s40519-016-0355-8

The introduction session needs to be improved to concentrate on hypotheses.

Methods

Page 3, line 108-109: There is a lack of information what kind of “ physical comorbidity” the participants presented?

In this section Authors should determine the response rate for EDs patients.

The reliability information of the EDI-2, YFAS 2.0 and CTQ among the current sample should be reported in this section.

Page 3, line 104: Please complete the number of the approval from the local ethics committee.

Results

Figure 1 should be described.  

In Figure 2, there is a lack of information about ß, SE and p.

I have a problem understanding the connection between Figure 1 and 2.

Discussion

In my opinion, a more detailed explanation of the findings should be presented. Authors should clarify why they results are important and justify undesired or unexpected results as well. What Authors consider to be their most significant or unanticipated findings? What are the practical implications of these findings?

Authors should provide some additional information what does this paper add to our knowledge about mentioned topic. In addition, I would propose to conclude the discussion section by making suggestions for further research.

Author Response

Reviewer 1:

In the present study, Authors mainly focus on retrospective childhood maltreatment, therefore I would improve the title by “The impact of retrospective childhood maltreatment on eating disorders as mediated by food addiction: a cross-sectional study”.

Response: We would like to thank the reviewer on this remark. It is important to make our readers able to differentiate between the assessments of concurrent vs past history of maltreatment on the participant’s eating disorder via food addiction. The title has now been changed accordingly.   

Introduction

In my opinion comparison between FA and ED (for showing the similarities and discrepancies) will definitely increase the value of the introduction. Perhaps Authors may wish to include some of interesting papers on FA in their introduction:

Gordon, E.L., Ariel-Donges, A.H., Bauman, V., Merlo, L.J. (2018). What is the evidence for “food addiction?” A systematic review. Nutrients, 10(4), 477. https://doi.org/10.3390/nu10040477.

Brewerton, T.D. (2017). Food addiction as a proxy for eating disorder and obesity severity, trauma history, PTSD symptoms, and comorbidity. Eating and Weight Disorders, 22(2), 241-247. https://doi.org/ 10.1007/s40519-016-0355-8

Response: We appreciate the reviewer’s advice. As a matter of fact, the interesting perspective published by Brewerton in 2017 has been already cited in the original version of our manuscript (page 2, line 61-65 and line 70-74). As for the systematic review done by Gordon et al., it has been cited in the modified version of our manuscript along with following:

In concordance with known suggested mediators of ED development in patients who have been exposed to childhood maltreatment, food addiction (FA) may constitute a yet unexplored contributing mediator. FA is characterized by poorly controlled intake of preferred foods which are postulated to act via similar mechanisms as both illicit and licit drugs of abuse in the brain [8]. An increasing amount of evidence of biological and behavioral changes in response to preferred foods (such as brain reward changes, impaired control, genetic susceptibility, substance sensitization and cross-sensitization, and impulsivity) have been sufficiently convincing to conceptualize FA as an addiction disorder [9]. FA has been increasingly considered as an important psychological dimension that leads, in patients with a history of complex trauma, to ED and more specifically binge eating disorders (BED) and bulimia nervosa (BN) [8]. Despite being a clinical manifestation of addiction to food, as much as 61.5% of patients with anorexia nervosa (AN) of the restrictive type have been found to suffer from FA which translates how much the addictive behavior on food can be a common pathological dimension to all ED as well as a possible accompanying manifestation of other forms of behavioral addiction to fasting, physical exercising, etc. [10].

The introduction session needs to be improved to concentrate on hypotheses.

Response: We agree with the reviewer that some of the introduction paragraphs were redundant and that the reader may not completely concentrate on the hypothesis while reading the original version of our manuscript. We have now eliminated a few redundant sentences, added some sentences and expressions and corrected some syntactical mistakes that may help the reader better understand our hypothesis.      

Methods

Page 3, line 108-109: There is a lack of information what kind of “physical comorbidity” the participants presented?

In this section Authors should determine the response rate for EDs patients.

Response: We agree on the fact that this sentence warrants further clarifications. We have modified it as follows:

Exclusion criteria were: Refusing to consent (n=3), having a mental disability such as intellectual deficiency (n=4) and having a physical comorbidity that prevented study participation (severe hypokalemia that necessitated a transfer to an intensive care unit n=4; very low nutritional state n=9; other physical disorders having an important secondary impact on cognitive functions n=5).

The reliability information of the EDI-2, YFAS 2.0 and CTQ among the current sample should be reported in this section.

Response: We have added the Cronbach’s alpha coefficient for internal consistency at the end of each paragraph describing the major assessment tools EDI-2, YFAS 2.0 and CTQ as requested by the reviewer.

Page 3, line 104: Please complete the number of the approval from the local ethics committee.

Response: In the original submitted version of our manuscript, we have mentioned the complete number of the approval from our local ethics committee as follows:

The data utilized here are drawn from a large study approved by the Ethics Committee of CPP Sud-Est VI of Clermont Ferrand University (CPP : AU 1313; ID-RCB : 2017-A00269-44; N° Clinical Trial : NCT03160443).

Results

Figure 1 should be described.  

In Figure 2, there is a lack of information about ßSE and p.

I have a problem understanding the connection between Figure 1 and 2.

Response: We thank the reviewer for all these important comments. We have been well-guided to realize that some ambiguity resided in the figures and their description in the text. As for the description of figure 1, we have modified the corresponding paragraph in the Results section as follows:

Findings from the mediation analyses are summarized in table 4 and significant effects are represented in figure 1. A direct effect between the CTQ subscales and the EDI-2 total score (unmediated by YFAS 2.0) was found for emotional and sexual abuse only (p=0.002 and 0.003 respectively). A consistent indirect mediation effect was present between all CTQ subscales and the EDI-2 total score via YFAS 2.0. The strongest indirect mediation effect was found in relation to the CTQ physical neglect subscale (standardized effect = 0.208; 95% CI [0.127-0.29]) followed by emotional abuse (standardized effect = 0.183; 95% CI [0.109-0.262]. Accordingly, the mediation or indirect effect of FA related to the impact of childhood maltreatment on clinical symptoms of ED seems to be more specific to physical neglect since, in addition to exerting the highest indirect effect among all CTQ subscales, it does not exert any direct effect on EDI-2 total score (Figure 2).

As for figure 2, SE corresponds to 95%CI. The ß coefficient of every effect corresponds to the “total”, “direct” and “indirect” unstandardized effect that also reflect the effect size. P values are not needed since they are implicitly reflected in 95%CI. We have opted for not avoiding mentioning the p values and SEs and ß in the figure in order to make it less confusing to the readers. However, some explanations have been added to the figure legend. Finally, we hope that the connection between both figures has been clarified throughout all these operated changes. 

Discussion

In my opinion, a more detailed explanation of the findings should be presented. Authors should clarify why they results are important and justify undesired or unexpected results as well. What Authors consider to be their most significant or unanticipated findings? What are the practical implications of these findings?

Authors should provide some additional information what does this paper add to our knowledge about mentioned topic. In addition, I would propose to conclude the discussion section by making suggestions for further research.

Response: We thank the reviewer for these important suggestions. We have modified the first paragraph of the discussion as follows:

“This study investigated the relationship among self-reported history of childhood maltreatment, FA and the ED symptom severity in a large sample of consecutively recruited patients with ED. Patients with FA reported more frequent histories of childhood maltreatment and presented with more severe ED symptoms as assessed by the EDI-2. In addition, the strong correlation between the YFAS and EDI-2 scores suggested that FA and ED symptom severity may be tightly related, while childhood maltreatment was less strongly related to FA. Emotional abuse seems to be the most important type of childhood maltreatment affecting ED symptoms severity. In addition, our findings revealed evidence of an indirect effect between all types of childhood maltreatment and ED symptoms severity via FA. The strongest of these indirect or mediated effects has been related to physical neglect followed by emotional abuse. Moreover, a direct effect relationship between childhood maltreatment and ED symptom severity emerged for the emotional and sexual abuse dimensions. Accordingly, our findings highlight the specific importance of FA in mediating the impact of physical neglect of the clinical severity of patients with any type of ED. Although cross-sectional, these findings are consistent with a model in which childhood maltreatment, especially physical neglect might precipitate or constitute a risk factor for FA which may later predispose for, maintain and/or exacerbate ED symptoms. The role of FA as a mediating factor of in the relationship between childhood maltreatment and ED warrants further exploration in longitudinal observational studies”.

In addition, in order for us to highlight our most important findings, we have started the paragraph related to the mediation effect of FA between physical neglect and eating disorder’s clinical severity with following sentence:

“Our most interesting finding is related to the fact that the strongest indirect effect of childhood maltreatment on ED symptoms severity via FA emerged for physical neglect”.

Finally, in relation to the summary of the major findings in our study and its implications for future research, we have amended the corresponding paragraph in the original version of our manuscript in order for it to better answer this specific reviewer’s remark. Accordingly, it became as follows:

“Patients with EDs and a history of childhood maltreatment may benefit from care that specifically targets this history [44]. Furthermore, the symptom pathway leading to an ED in individuals with a history of childhood maltreatment has been described as specific to this group. Thus, overvaluation of weight and body shape may lead to feelings of loss of control followed by depressive symptoms and, subsequently overeating [45]. Tailoring of usual treatment protocols to account for these pathways may help to improve clinical outcomes. Given the evidence found in this study for the mediating role of FA, it would be interesting to evaluate whether, in addition to the usual treatment, therapeutic strategies specifically targeting FA might improve overall prognosis. The presence of a history of physical neglect should raise the clinicians’ index of suspicion for the presence of FA. In this regard, in case FA has been confirmed as a comorbid clinical entity accompanying the ED (after having used screening tools such as YFAS), the treatment of FA clinical dimensions and its overall impact on ED symptoms should be assessed in future studies. Accordingly, known suggested treatment protocols such as combining pharmacotherapies (opiate antagonists) and psychotherapies such as cognitive behavioral therapy and psychodynamic group treatments may be successful in targeting FA clinical dimensions and subsequently ED symptoms severity [46, 47]”.

As for the conclusion, we have also made some modifications in order for it to answer the reviewer’s remark:

“In conclusion, although cross-sectional, our findings support the existence of a mediated relationship between childhood maltreatment and ED, via FA, especially in the presence of a history of physical neglect. Patients with severe ED symptoms and a history of childhood maltreatment should be systematically assessed for the presence of FA. Moreover, when childhood maltreatment is documented in patients with ED, tailoring treatment plans to specifically address FA should be considered.

Reviewer 2 Report

The authors have undertaken an analysis of an extremely important problem, which is the assessment of the relationship between child maltreatment and the occurrence of eating disorders later in life. An interesting approach is an attempt to determine the place of food addiction in this relationship.

In general, the article is very well-written. Both the eating disorders and food addiction are presented in the introduction to the article. This part of the paper also contains a description of the basic assumptions of the paper, which were formulated in a clear and unquestionable way.

The following parts of the manuscript also do not raise any major objections. Both the description of the participants, the way they were recruited, ethics rules, the tools used and the planned methods of analysis are presented in a transparent way. Moreover, the methods of analysis were chosen correctly to the announced research assumptions.

The results are described in a reliable manner and with attention to detail, Similarly, the tabular and graphic presentation of the results is friendly to the reader's eye.

The proposed discussion is exhaustive and fully responds to the research problem. The authors write with full awareness of the limitations resulting from the research undertaken, which proves their research maturity and reliability.

In my opinion, the manuscript presented by authors should be recommended for publication after minor corrections. I kindly ask the authors to take into account the following problems:

1 Abstract

 - please write down the age of the participants and the period of the research.

  1. Introduction:

 - the parts presented in 2 paragraphs in lines 61-69 and 70-81 are very similar. I ask you to verify these contents,

- please add a short paragraph to define the child maltreatment phenomenon.

  1. Materials and methods:

- participants - please complete the information about the age range of participants (Me of age is presented in the table),

- in Table 1 the authors present the value of the BMI - please complete the information on how body weight and height data were collected by the researchers (anthropometric measurements at the doctor’s office or self-reported data).

  1. Statistical analysis:

- Please complete the information on how the BMI variable was used in addition to the analyses presented in Table 1.

Round 2

Reviewer 1 Report

I am satisfied with the revision made on the manuscript. I only have three remarks:

1. In whole manuscript (not only in its title) “RETROSPECTIVE childhood maltreatment” should be changed (e.g. “we therefore hypothesized that childhood maltreatment…”)

2. Line 66-68, page 2: The sentence “In concordance with known suggested mediators of ED development in patients who have been exposed to childhood maltreatment, food addiction (FA) may constitute a yet unexplored  contributing mediator” needs to be explained. What “known suggested mediators of ED development” Authors have in mind? In my opinion, this sentence is too general and does not contribute new knowledge.

3. Methods: The reliability information of the EDI-2 should be improved as following: “Cronbach’s alpha coefficient for internal consistency ranged from …. to …….. for EDI-2 subscales”. Authors used all subscales of the EDI not only the EDI total score.
